# Performance Analysis of Routable GOOSE Security Algorithm for Substation Communication through Public Internet Network

**DOI:** 10.3390/s23125396

**Published:** 2023-06-07

**Authors:** Soohyun Shin, Hyosik Yang

**Affiliations:** Department of Computer Science and Engineering, Sejong University, 209, Neungdong-ro, Gwangjin-gu, Seoul 05006, Republic of Korea; schz307@naver.com

**Keywords:** micro-grid, IEC 61850 routable GOOSE protocol, data aggregator, security, public internet network

## Abstract

Traditional unidirectional power systems that produce large-scale electricity and supply it using an ultra-high voltage power grid are changing globally to increase efficiency. Current substations’ protection relays rely only on internal substation data, where they are located, to detect changes. However, to more accurately detect changes in the system, various data from several external substations, including micro-grids, are required. As such, communication technology regarding data acquisition has become an essential function for next-generation substations. Data aggregators that use the GOOSE protocol to collect data inside substations in real-time have been developed, but data acquisition from external substations is challenging in terms of cost and security, so only internal substation data are used. This paper proposes the acquisition of data from external substations by applying security to R-GOOSE, defined in the IEC 61850 standard, over a public internet network. This paper also develops a data aggregator based on R-GOOSE, showing data acquisition results.

## 1. Introduction

Existing power systems that generate, transmit, distribute, and consume power on a large scale have problems concerning efficiency due to various factors. In situations where bidirectional communication is required to increase efficiency, micro-grids and smart grids use various kinds of renewable energy, and a small stand-alone power grid is typically used as a solution [1,2,3,4]. Similar efficiency problems exist in substation environments. Current substations’ protective relays detect changes using only internal data from an internal substation where they are located and perform functions such as blocking circuits in the event of a short circuit or in the event of abnormal conditions in an electrical circuit. However, the current protection relays determine the setting value of the individual protection relay, so it cannot reflect changes in the state of the power grid. Relying solely on data from an internal substation may cause power accidents in special situations, such as disaster situations, and data from several external substations, including micro-grids, are required to detect more accurate system changes.

To solve this problem, a data aggregator that acquires data in real-time has been developed. The data aggregator serves to acquire data from adjacent substations with centralized smart protection technologies that can centrally protect substations using dynamic state estimation results [5]. Data must be obtained from the intelligent electronic devices (IEDs) in internal substations and external substations in real-time, but data acquisition from external substations located a great distance away is limited in cost and security and is only now starting to use the GOOSE (generic object-oriented substation events) protocol defined in the IEC 61850 standard. Data acquisition from distant external substations can lead to the destruction of the power system through cyber attacks [6,7,8,9], thus requiring secure communication. However, building a separate private network to communicate with external substations, including most micro-grids on islands and mountains, is costly.

International standards have been enacted for communication between substations, but most of them rely on private protocols from IED vendors. Using private protocols increases system complexity and interoperability, but protocols such as MODBUS and DNP3 are still used in the industry [10]. To overcome the limitations of these private protocols, IEC 61850, an international standard for power automation that defines data models and communication methods, has been developed [11,12,13,14].

Recently, various studies have been conducted on communication using R-GOOSE (routable GOOSE) defined in IEC 61850 90-5 and on substation communication to which security is applied [15,16,17,18]. Among them, an experiment using R-GOOSE as a multicast communication framework for communication between substations was conducted at Florida International University (FIU). GOOSE data models were encapsulated in data distribution service (DDS) data objects and were routed to final destinations over the network, and the actual hardware-based tests developed using FIU’s smart grid testbeds, in terms of end-to-end latency, showed that they met the maximum time range defined in the IEC 61850 standard [19]. However, the experiment was conducted on a private network of the FIU’s testbed and factors such as network hops and CPU load on a public internet network were not taken into account. There was also a study on the definition of R-GOOSE which also presented an application proposal of the differences from existing GOOSE messages, power distribution automation, and protection systems [20]. Other studies include Jaya R. A. K. Yellajosula’s implementation of control between substations using R-GOOSE, the allowable range of PURTT (permissive under reach transfer trip), and an interlocking application between two substations. However, this study also used a private internet network, not a public internet network, and no research on security was conducted [21]. Research on cyber attacks on IEC 61850-based networks has been conducted [22], and research on security threats presented in the IEC 62351 standard for message exchange has also been conducted, but no research has been conducted to verify communication performance [16]. Communication performance is another important metric in the smart grid. Recent research shows that IoT (Internet of Things) protocols, e.g., DDS and XMPP (extensible messaging and presence protocol) can meet the timing constraints defined in IEC 61850-90-5 [23,24]. These studies, however, used a data model and not a service model, as defined in IEC 61850, due to the lack of mapping between IoT protocols and IEC 61850. This research shows that the IEC service model, e.g., R-GOOSE, can be used directly on a public internet network while satisfying timing constraints.

Research on R-GOOSE security was conducted with the implementation of MAC (message authentication code) algorithms, such as HMAC (hashed message authentication code) and AES (advanced encryption standard)-GCM (Galois counter mode) integrity, showing that the application of encryption algorithms is suitable for communication satisfaction, but no research has been conducted on confidentiality where a private internet network was used [25].

This paper deals with the acquisition of external substation data through a public internet network using R-GOOSE equipped with a security algorithm that complies with IEC 61850, an international power automation standard, to ensure interoperability. Based on the data acquisition results, a secured R-GOOSE-based data aggregator was developed, and the communication performance between substations through a public internet network was analyzed. To the best of our knowledge, this is the first relative performance analysis of R-GOOSE on a public internet network. Furthermore, this paper performs a performance comparison of each security algorithm employed, as well as a comparison of the system performance that takes into account factors such as CPU load and packet size.

## 2. Data Aggregator Structure

The data aggregator serves to acquire data from adjacent substations among centralized smart protection technologies that can centrally protect substations using dynamic state estimates. Real-time data (voltage, current) are received from smart protection IEDs of internal and external substations in the form of GOOSE or R-GOOSE, and protection is performed by exchanging voltage, current, and distance relay elements.

Data aggregators must acquire data from external substations in real time, but external communication without using a private internet network faces security and cost limitations. Due to this problem, only internal data are currently acquired using GOOSE from the IEDs of internal substations. This satisfies security and cost issues, and even if a new power grid is added, data must be acquired using a public internet network to acquire data from an external substation without building a separate private internet network.

The data aggregator models, namely, Logical Devices (LDs) and Logical Nodes (LNs) defined in the actual IEDs in the data aggregator, are treated as one IED and expressed as objects in the SCL file. The data aggregator must have an external data communication function with an external substation, including a micro-grid, and since the existing power system is operated by establishing and operating a private internet network, external communication is impossible.

### 2.1. Communication between Substations

The communication methods between substations defined in the IEC 61850 90-1 document are largely divided into two methods: tunneling and proxy/gateway. The data aggregator developed in this paper was adopted because the proxy/gateway method was more advantageous because it replaces the data inside the substation like a proxy. The proxy/gateway method places a proxy in a substation that is accessed through a proxy in an external substation. Inside the substation, it acts like a client to acquire data from a single substation in real time, and in an external substation, it acts like a server to transmit data to the proxy of an external substation. The proxy may mirror the LD in another IEC 61850 physical device (PHD) and build a proxy/gateway using the LD. As shown in Figure 1 below, proxy B2 provides a subset of information necessary for A2, from the perspective of A2. External substations may provide other functions for efficient use of communication mechanisms. For example, in the case of GOOSE, only state changes are actually transmitted, retransmissions with the same state information may be filtered from the transmitter, and retransmissions may be locally re-generated in the proxy. Subsequently, the retransmission missing from the transmitting side must send a signal to the proxy through the state information of the external substation [25].

The following Figure 2 shows how the PHD is mapped to an LD serving as a proxy gateway. The A.LD1 and ASYS of LD are copied to the proxy/gateway, and the LPHD of A.LD1 in the proxy/gateway represents PHD.A. The LD may also include a domain-specific LN that corresponds to an external adjacent substation and in turn corresponds to PHD.C. Furthermore, data provided by PHD.C are mapped to C.LD2. It can be seen that the external physical device is mapped to the proxy/gateway device [26]. R-GOOSE, which the data aggregator will use to communicate externally, also operates in a server/client manner, and the LN on each IED is released to the external substation instead as the LN inside the data aggregator.

The data aggregator developed in this paper also has multiple LDs in one IED and consists of multiple LNs in the LD. Each LD maps to each LD as IED1.LD1, and each LD has LNs such as LPHD and LLN0 that provide information for each LD. Because the data inside the data aggregator represent the information reflected in the data aggregator rather than the actual IED value, the Proxy DO (Data Object) value in the LPHD should be displayed. LD1.LPHD.Proxy in LD’s LPHD LN, which reflects the actual IED value in the data aggregator, is set to TRUE. In LPHD in the actual IED, all proxies are set to FALSE, and LD0.LPHD.Proxy is also set to FALSE.

### 2.2. Communication Requirements between Substations

The IEC 61850 90-5 document defines the requirements for synchronous phasor data communication. Requirements include transmission speed, delay time from measurement to application, delay variation, and transmitted reliability. Table 1 summarizes the requirements. In principle, the Sampled Value service applies to all applications, but cannot be used in the WAN (Wide Area Network) due to direct mapping to Ethernet. Therefore, a routable Sample Value service and R-GOOSE may be used [27].

In addition, the IEC 61850 90-1 document also defines communication requirements. Regarding message performance, it is defined by dividing the message required for control and protection and the message required for power quality and measurement. The data aggregator deals with the messages needed for control and protection. The requirements for the communication performance of control and protection are basically the same between one bay and another and between substations. Therefore, the same classification scheme is used for all connections that meet the IEC 61850 standard. The automation function for the interaction between the IED and the IED has two conditions. First, in the case of a high-speed state-based application, the transmission time should not exceed 20 ms. Second, for steady-state-based applications, the transmission time may exceed 20 ms but should not exceed 100 ms, with an upper limit of 100 ms [28]. Therefore, in this study, 20 ms was used as the standard for communication performance based on two documents.

### 2.3. IEC 61850 Analysis for Data Aggregator Modeling

Currently, there are various private protocols for communication. Using all of these may cause an increase in system complexity, and since they are not optimized for substations, additional data modeling was required. There was also a problem that the system was not suitable for high-speed communication. In addition, there was a problem with the interoperability due to different private solutions used by IED vendors, and to address this, it was necessary to define a data communication protocol commonly used by IED vendors. The IEC 61850 standard provides object-oriented data and service modeling to address interoperability and optimize substation automation as a single international standard.

As shown in Figure 3, IEC 61850-7-2 defines a data class, including attributes and services; IEC 61850-7-3 defines a CDC (common data class) inherited by a common attribute; and IEC 61850-7-4 defines a DO that constitutes an LN using a CDC [27]. The physical IED mapping of the data aggregator virtualized the LD and LN defined in the physical IED, which is one of the core concepts of IEC 61850. The data aggregator should be treated like an IED and represented as an object within the SCL (System Configuration Language) file.

The connection between the data aggregator that acquires the internal data of the substation and the actual IED is shown in Figure 4. Data on PTOC, which is the LN corresponding to protection, are transmitted using GOOSE messages, and the received data can be mapped to each LN and DO and stored in the database and used later.

## 3. Routable GOOSE with a Security Algorithm for Public Networks

### 3.1. Routable GOOSE

R-GOOSE, which ensures interoperability and that the system can communicate externally, is defined in the IEC 61850 90-5 and IEC 61850 8-1 documents. The control and configuration service uses the existing IEC 61850 standard method along with MMS over TCP/IP and does not require a separate extension. In the case of data transmission, a new UDP (User Datagram Protocol) mapping is required, but GOOSE for current internal substation communication is also proposed in the standard, allowing the tunneling of existing Ethernet-bound GOOSE packets via UDP/IP [29].

R-GOOSE operates in a server/client manner, as defined in the IEC 61850 90-5 document, and communicates with external substations through a private or public internet network, unlike GOOSE, which operates on the data-link layer and can only transmit and receive data inside the substation subnet using a MAC address [27].

### 3.2. Security Algorithm for R-GOOSE

Data communication through a public internet network has the disadvantage of being vulnerable to security threats, as defined in the IEC 62351-1 and IEC 62351-2 documents, so data aggregators, which acquire data from external substations through the public internet network, must be equipped with strong security technology. The IEC 61850 90-5 document defines the security of R-GOOSE. Confidentiality is defined as optional, and information authentication and integrity are defined as mandatory. In theory, authentication and integrity are provided in an end-to-end manner regardless of the information hierarchy and generally provide security features using a MAC address. Hardware performance must be considered to be equipped with security. Previously, IEC 62351-6 documents used digital signatures using asymmetric encryption algorithms, but problems occurred due to slow downs in terms of hardware cost and CPU performance. IEC 61850 90-5 defines an encryption algorithm in consideration of this problem. In addition, various security requirements and traffic classes were divided according to the speed and type of message transmission, and the document defined requirements suitable for the two classes. Based on the security requirements defined in the class, the documents define means of implementing authentication and integrity, and the basic security structure also ensures the integrity of the selected security mechanism on an end-to-end basis. Moreover, the IEC 61850 90-5 document defines an HMAC using SHA-256, which generates hash values for integrity and authentication, and an advanced encryption standard allows message authentication code AES-GCM to be used. Although confidentiality is optional, AES-128 and AES-256 for R-GOOSE and routable Sampled Value messages are specified as encryption algorithms [30].

In this paper, communication performance was verified by developing R-GOOSE and applied by combining HMAC-SHA256 and AES-128, which are security algorithms specified in the standard, and r-GOOSE applied AES-128-GCM.

### 3.3. Network Layer Analysis for Using Security Algorithm

Among the network layers defined in the standard, the session layer consists of security-related header fields, as shown in Figure 5. Packets generated in the session layer consist of the SI (session identifier), followed by the length of all parameter fields in the session header, except for the user information field.

The SPDU (Session Protocol Data Unit) length is 4 bytes and consists of the total length from the SPDU number to the HMAC field. The SPDU number represents a unique identification of session packets that 4 bytes in size and detects the replication of packets on the target device [14]. According to the IEC 61850-90-5 document, SI (Session Identifier) has four possible values. In addition, the PV (Protocol Version) consists of the SPDU length, the SPDU number, the version number, the time of the current key, the time of the next key, the security algorithm, and the key ID.

## 4. Performance Analysis of Routable GOOSE

To analyze the performance of R-GOOSE, testbeds with a data aggregator, R-GOOSE publisher, and subscriber are modeled. The testbed specifications used in the analysis are shown in Table 2.

### 4.1. R-GOOSE Communication Test Environment via a Public Internet Network

As R-GOOSE is defined in the IEC 61850 90-5 document, the MMS-lite Version 6.2 API supports R-GOOSE. The GSEControl section in the SCL file must be changed to use R-GOOSE. The SCL file should indicate the use of R-GOOSE, such as the protocol in the GSEControl section; the R-GOOSE publisher should define the subscriber in IEDName; and the publisher and subscriber should use the same SCL file. R-GOOSE communicates using the multicast UDP protocol, so the addressing method is different. In the address section, an IP-IGMPv3Src (Internet Group Message Protocol) address must be defined, along with the address to join each other’s promised groups. The IP address must be a valid multicast IP address, and the IP-IGMPv3Src address can be any valid IP address, which is sent from R-GOOSE to the source address, and the IED source address is sent if there is no IP-IGMPv3Src in the address section.

In order to verify that the external substation can communicate normally with the external substation using the R-GOOSE, an external communication test using the R-GOOSE was conducted in a remote IED-to-IED public internet network environment, and RTT (round trip time) was measured to analyze whether the communication requirements defined in the standard document were met.

As shown in Figure 6, an IED acting as a server (developed in the laboratory of Sejong University), an IED serving as a client in a virtual machine, and the two IEDs were connected in a public internet network. The data transmitted from Sejong University’s server IED with an LN called MMXU2 was received by the client IED located in the virtual machine, and the received data were retransmitted to the server IED to measure the RTT. The RTT was calculated by dividing the transmitted time and the retransmitted time by 2, and a total of 1 million data were used as sample data.

The test used 1 million sample data. The test results showed an average value of 24.748 ms, a maximum value of 1042 ms, and a minimum value of 10 ms, as well as a loss rate of 0.014%, as shown in Table 3. It can be seen that the average 24.748 ms RTT satisfies the 20 ms requirement based on communication performance in this study. Through this, it was shown that R-GOOSE can be used through a public internet network.

### 4.2. Performance Analysis of R-GOOSE via a Public Internet Network with a Security Algorithm

In an environment such as the test in Section 4.1, a test was conducted to verify that R-GOOSE equipped with a security algorithm that combines HMAC-SHA256 with AES-128 to ensure the authentication, integrity, and confidentiality defined in standard documents meets the communication requirements. The test was organized as shown in Figure 7 below, and the process was divided into six steps for measurement. The first process is to encrypt the plaintext using the AES-128 security algorithm in Sejong University’s lab IED and then generate an HMAC that uses HMAC-SHA256 using ciphertext. Thereafter, the ciphertext and HMAC are transmitted to the Vultr Seoul virtual server IED as R-GOOSE through the public internet network. The Vultr virtual server IED retransmits the received ciphertext and HMAC to the Sejong University lab IED. The Sejong University IED generates HMAC’ using the ciphertext received from Vultr, and then compares the HMAC that was initially generated and retransmitted to check the integrity and proceed with the ciphertext decryption process. The timestamps of the six processes were checked to obtain the round trip time.

In an environment such as the experiment in Section 4.1, an experiment was conducted to verify that R-GOOSE with improved security satisfies the communication requirements by installing an AES-128-GCM security algorithm that can guarantee three elements of security outlined in standard documents. The experiment was organized as shown in Figure 8 below, and the process was divided into three steps for measurement. In the first process, the plaintext is encrypted using the AES-128-GCM security algorithm at Sejong University’s lab IED, and then the ciphertext is transmitted to the virtual machine IED via the public internet network. The Vultual machine IED retransmits the received ciphertext to the Sejong University lab IED. The Sejong University IED decrypts the ciphertext received from the virtual machine. The round trip time is obtained using the timestamp for three processes.

The test was conducted 10 times using 1 million data as a sample, and the test was conducted at 10 a.m. and 5 p.m., respectively, in consideration of the load on the network by time zone. When applying the HMAC-SHA256 and AES-128 algorithms, the total round trip time averaged 41.9607 at 10 a.m. and 41.9155 at 5 p.m. Generating and verifying the HMAC resulted in an average time of 9.4305, while encryption and decryption showed an average time of 2.9317. When applying the AES-128-GCM algorithm, the total round trip time averaged 39.0426 at 10 a.m. and 38.1552 at 5 p.m. Encryption and decryption showed an average time of 6.5592. With regard to the RTT, the processing time for the security algorithm does not affect the RTT critically. That is, as shown in Table 4, the time of the day does not affect the total RTT.

Different packet sizes for R-GOOSE were applied. The two security algorithms were applied, and the test was conducted in the same way using a total of three packet sizes: 100 bytes, 213 bytes, and 426 bytes, depending on different contents of GOOSE messages. The test was conducted 10 times using 1 million data as a sample. The average total time taken to apply the HMAC-SHA256 and AES-128 algorithms was 12.3035 for a packet size of 100 bytes, 12.654 for a packet size of 213 bytes, and 12.8806 for a packet size of 426 bytes. On the other hand, when applying the AES-128-GCM algorithm, the average time was 6.734 for a packet size of 100 bytes, 6.8645 for a packet size of 213 bytes, and 6.9346 for a packet size of 426 bytes. The test results are shown in Table 5 below and show that packet size does not critically affect the processing time.

We also analyze the CPU load of the security algorithms. We run the six IED processes on one machine, where each IED process takes up 15% of the CPU, and measure the processing time of each security algorithm. The test was conducted 10 times using 1 million data as a sample. The average time required for processing six IEDs using the HMAC-SHA256 and AES-128 algorithms was 13.6271 ms, and when applying the AES-128-GCM algorithm, the average time for processing six IEDs was 6.908 ms. In comparison with Table 4, the security algorithm does not take much CPU power, and there is not much difference from the simultaneous process, as shown in Table 6.

Through this test, it was found that among the security algorithms that satisfy the three elements defined in the standard document, the AES-128-GCM algorithm performed better than HMAC-SHA256 + AES-128. In addition, it was found that the CPU load and packet size of the IED did not have a significant impact on communication between substations and satisfied the communication performance standard of 20 ms.

### 4.3. Data Aggregator Data Acquisition

The internal and external substation data acquisition experiment testbeds using R-GOOSE equipped with GOOSE and security functions in a public internet network environment were constructed, as shown in Figure 9. The test procedure is to transmit GOOSE located in the internal substation network environment with distribution line protection, transmission line protection, and bus protection and R-GOOSE with security technology from a central protection IED located in the Vultr virtual server and built in the external substation network environment. Since four IEDs located inside and outside need to communicate with the data aggregator, a network was constructed using switches and routers. Three IEDs were connected to Sejong University’s private internet network that transmits GOOSE and is connected to a switch and communicates using a MAC address. Secure R-GOOSE configured the network using a router to communicate with the data aggregator using a fixed IP.

Figure 10 shows a captured screenshot that simultaneously acquires experimental data from the internal and external substation environments of the data aggregator experimented in this section. As a result of the test, it was shown that data could be obtained from four IEDs normally.

The following Figure 11 shows a screenshot of the secured R-GOOSE packet captured using Wireshark.

## 5. Conclusions

IEC 61850, the international standard for power system automation, defines the data and service model. The first edition of IEC 61850 defines the GOOSE service for timely delivery of commands and data. Due to the lack of mapping of the transport layer and network layer, communication inside the same subnet was possible. Thanks to R-GOOSE, communication outside of the substation, which means communication between different subnets, was achievable. This is critical for the protection of transmission lines, communication between substations, and communication between substations and the control center, which are new parts of the IEC 61850 series of standards. This paper presents a feasibility study of the use of R-GOOSE over a public internet network. This study defines the need to acquire data from external substations through a public internet network and develops a data aggregator that complies with IEC 61850 and acquires data from external substations with security technology. A performance analysis was conducted in an external substation environment using R-GOOSE messages without security technology and R-GOOSE messages with security technology, and the RTTs from transmission to transmission were measured. Even with security provision, i.e., AES-128 GCM, the average RTT between substations was measured in the tens of micro-seconds range. From the performance analysis, we verify that the R-GOOSE satisfies the timing requirements defined in [28].

In addition, since the R-GOOSE messages are transferred using the multicast UDP protocol, which has no acknowledgment function, it cannot guarantee the delivery of messages, which may cause the possibility of losing data in an environment where abnormalities such as short circuits from external substations must be transmitted or acquired. Therefore, it was confirmed that it is suitable for use in a data aggregator used in a protective relay that relies only on data from internal substations and requires data from external substations such as micro-grids located on islands and mountains.

In the future, it will be possible to conduct research on the design of external communications with improved security in a micro-grid environment distributed on islands and mountains using R-GOOSE to which a security algorithm is applied, and this implementation is expected to keep pace with changes in the global power market. As the penetration of IoT protocols, especially in micro-grids, increases, performance comparisons between R-GOOSE and IoT protocols, i.e., DDS and XMPP, should be conducted. 

## Figures and Tables

**Figure 1 sensors-23-05396-f001:**
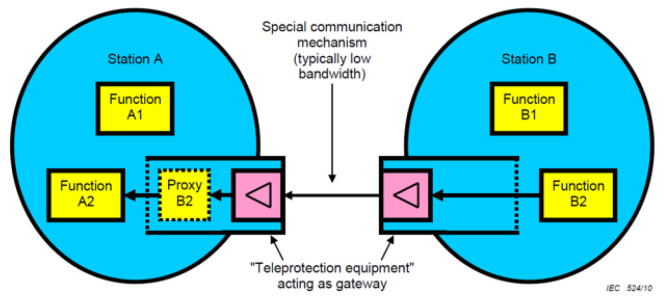
Communication structure using Proxy/Gateway [26].

**Figure 2 sensors-23-05396-f002:**
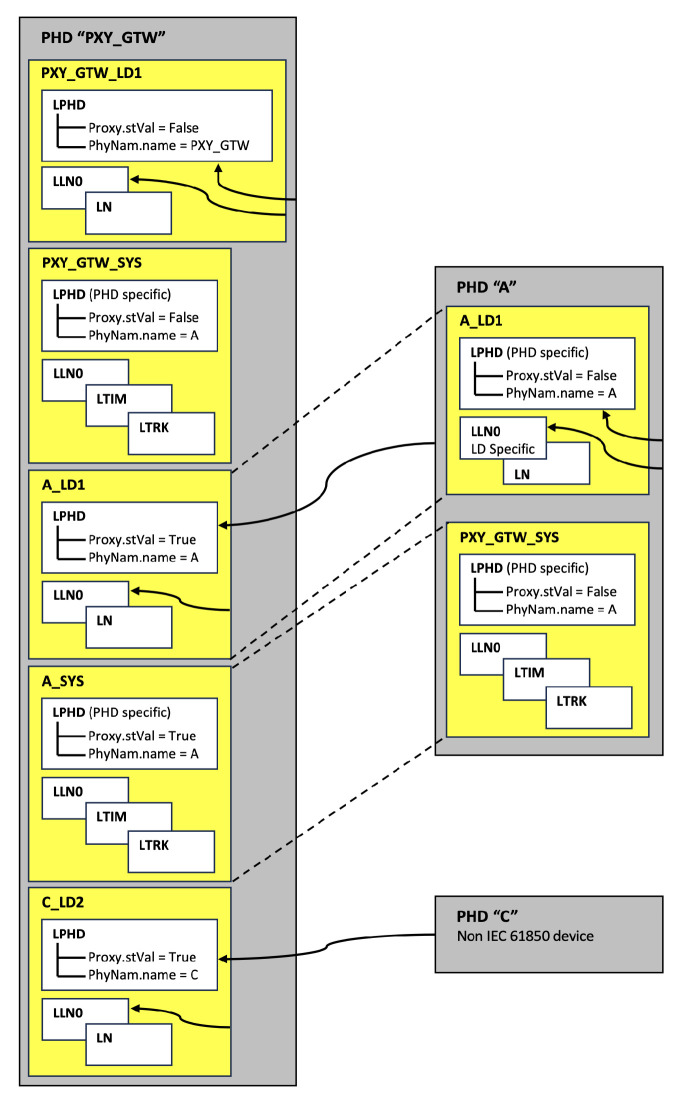
LogicalDevice in Proxy/Gateway [26].

**Figure 3 sensors-23-05396-f003:**
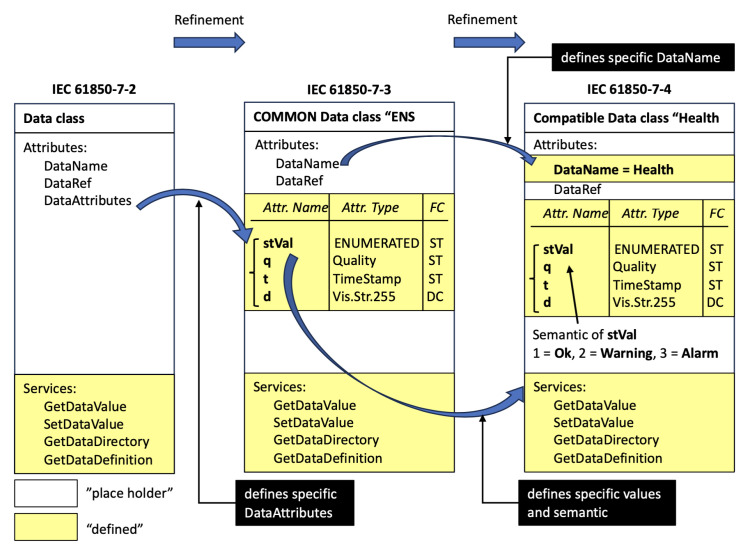
IEC 61850 Data Class Mapping [26].

**Figure 4 sensors-23-05396-f004:**
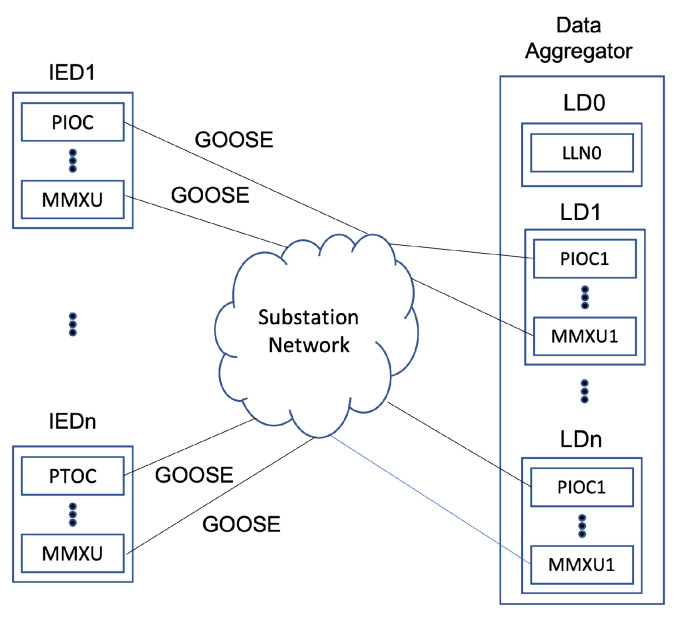
Data aggregator internal mapping.

**Figure 5 sensors-23-05396-f005:**
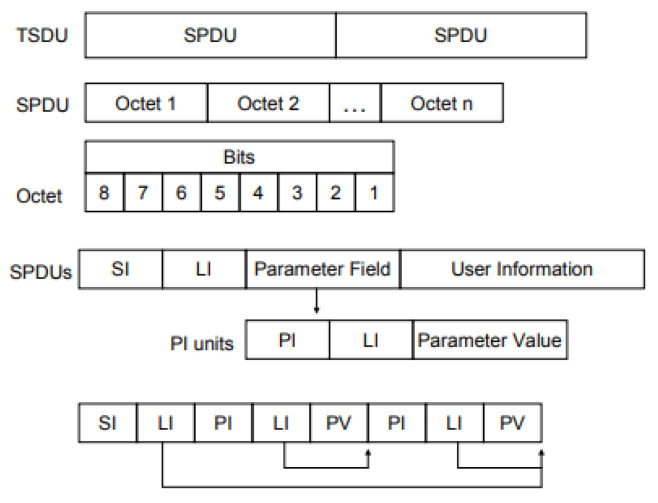
Session protocol header field [30].

**Figure 6 sensors-23-05396-f006:**
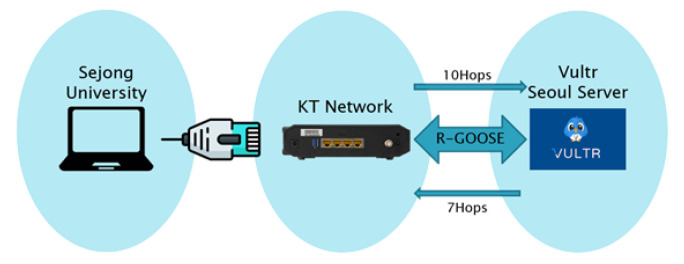
R-GOOSE RTT measurement over a public internet network.

**Figure 7 sensors-23-05396-f007:**
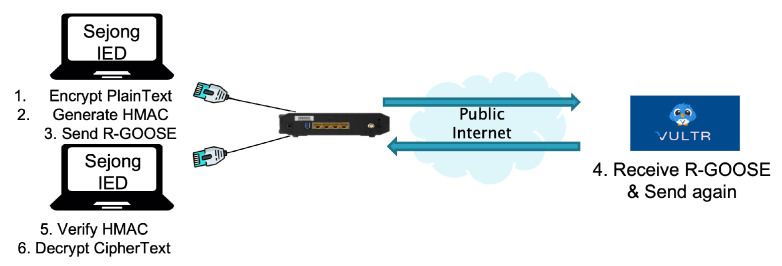
Performance analysis of secured R-GOOSE with HMAC-SHA256 + AES-128.

**Figure 8 sensors-23-05396-f008:**
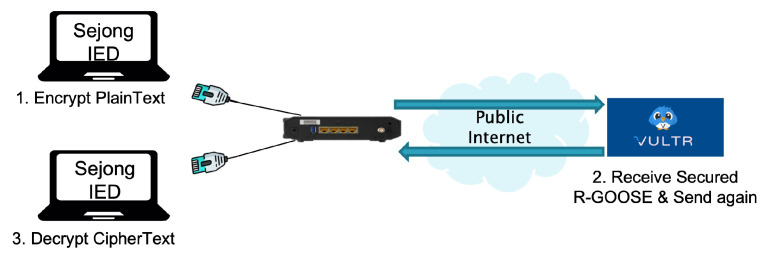
Performance analysis of R-GOOSE secured with AES-128-GCM.

**Figure 9 sensors-23-05396-f009:**
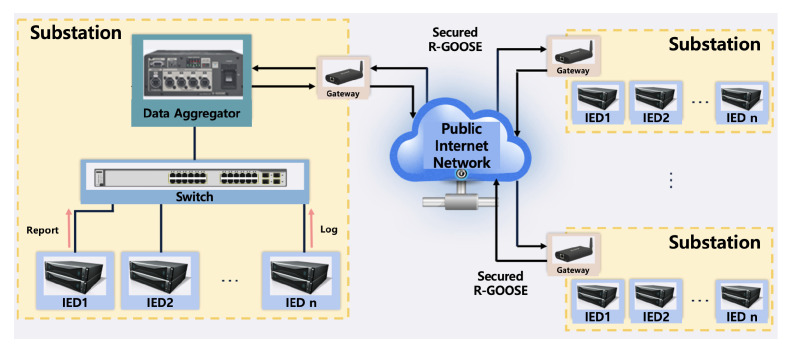
System configuration of data acquisition using R-GOOSE.

**Figure 10 sensors-23-05396-f010:**
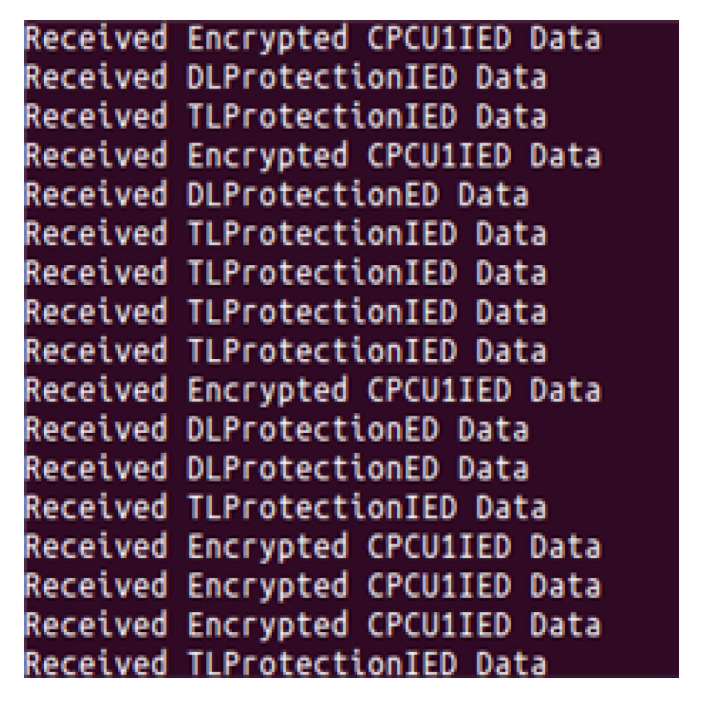
Screenshot of the data acquisition process.

**Figure 11 sensors-23-05396-f011:**
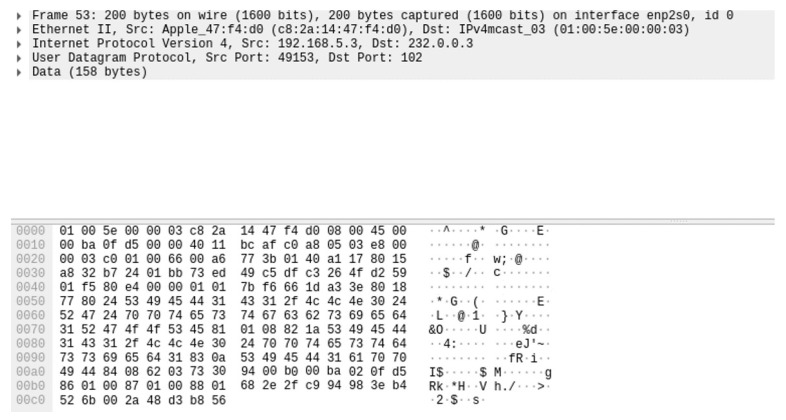
Screenshot of R-GOOSE packet.

**Table 1 sensors-23-05396-t001:** Communication requirements.

Factor	Reporting Rate Range	End-to- End Latency	Measurement Timing Error	Sensitivity to Lossy Packets
Sync-check	≥4/s	100 ms	50 μs	High
Situational awareness	1/s to 50/s	5 s	50 μs	Low to Medium
Wide area controls	≥10/s	50 ms to 500 ms	50 μs	High
PMU to PDC	1/s to 10/s	5 s	50 μs	Low to Medium
PMU to IEC	50/s or 60/s	20 ms	50 μs	High

**Table 2 sensors-23-05396-t002:** Specification of testbeds.

Role	Host	Hardware Specification			Software
		CPU	RAM	HDD	OS
GOOSE publisher	PC	Inter Core i7 3.6 GHz	8 GB	256 GB	Windows 10
Secured R-GOOSE publisher IED	Virtual machine	Core 2.4 GHz	1 GB	256 GB	Ubuntu 20.04.2
Data aggregator	Laptop	Intel Core i3 2.4 GHz	4 GB	500 GB	Ubuntu 16.04

**Table 3 sensors-23-05396-t003:** Round trip time of R-GOOSE measured in Figure 6.

**Total RTT**	**Average**	**Max**	**Min**	**Loss**
	24.748 ms	1042 ms	10 ms	0.014%

**Table 4 sensors-23-05396-t004:** Measured RTT of R-GOOSE with security algorithm.

**HMAC-SHA256 + AES-128**			
(ms)	10 a.m.	5 p.m.	Average
Total RTT	41.9607	41.9155	41.9381
Maximum RTT	1894	2208	
Send to receive RTT	29.8055	29.3462	29.5758
Generate HMAC	4.3756	4.7785	4.427
Verify HAMC	4.8914	5.1156	5.0035
Encrypt plaintext	1.4934	1.4246	1.459
Decrypt ciphertext	1.3948	1.5506	1.4727
**AES-128-GCM**			
(ms)	10 a.m.	5 p.m.	Average
Total RTT	39.0426	38.1552	38.5989
Maximum RTT	1107.71	1103.98	
Send to receive RTT	32.3095	31.7697	32.0396
Encrypt plaintext	3.5059	3.2264	3.3661
Decrypt ciphertext	3.2272	3.159	3.1931

**Table 5 sensors-23-05396-t005:** Performance analysis of R-GOOSE with different packet sizes.

**HMAC-SHA256 + AES-128**			
(ms)	100 byte	213 byte	426 byte
Total time	12.3035	12.654	12.8806
Generate HMAC	4.4815	4.5327	4.6124
Verify HAMC	5.0031	5.2411	5.3186
Encrypt plaintext	1.4083	1.4373	1.4677
Decrypt ciphertext	1.4106	1.4429	1.4819
**AES-128-GCM**			
(ms)	100 byte	213 byte	426 byte
Total time	6.734	6.8645	6.9346
Encrypt plaintext	3.3902	3.4459	3.4615
Decrypt ciphertext	3.3438	3.4186	3.4731

**Table 6 sensors-23-05396-t006:** Performance analysis of R-GOOSE with multiple processes.

**HMAC-SHA256 + AES-128**		
Total processing time (ms)	Average	IED’s Average
No.1 IED	13.4893	13.6271
No.2 IED	13.4788	
No.3 IED	13.5305	
No.4 IED	13.5512	
No.5 IED	13.8326	
No.6 IED	13.8803	
**AES-128-GCM**		
Total processing time (ms)	Average	IED’s Average
No.1 IED	6.7984	6.908
No.2 IED	6.7847	
No.3 IED	6.8423	
No.4 IED	6.9221	
No.5 IED	7.0483	
No.6 IED	7.0524	

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
