# Peer review of "Performance Analysis of Routable GOOSE Security Algorithm for Substation Communication through Public Internet Network"

_sensors, 2023, doi:10.3390/s23125396_

Round 1
Reviewer 1 Report
The study is relevant. Presented research deals with developing data aggregators based on Routable GOOSE. The manuscript can be published after a little bit improving. Firstly, need to add the list of the abbreviatures. Also, will be useful to revise the practical benefits of using Routable GOOSE.
The study is relevant. Presented research deals with developing data aggregators based on Routable GOOSE. The manuscript can be published after a little bit improving. Firstly, need to add the list of the abbreviatures. Also, will be useful to revise the practical benefits of using Routable GOOSE.
Author Response
First of all, we wish to thank all the reviewers for their constructive comments. These comments helped us to improve the quality of our manuscript. All the issues raised during the review process were addressed in this revised version of the manuscript, hopefully in the most satisfactory way and highlighted in red pen inside the paper. In the following lines, we detail the changes we have made according to the reviewers’ comments.
Point 1: The study is relevant. Presented research deals with developing data aggregators based on Routable GOOSE. The manuscript can be published after a little bit improving. Firstly, need to add the list of the abbreviatures. Also, will be useful to revise the practical benefits of using Routable GOOSE.
In this revised version of our manuscript, we include the abbreviation part.
We add the following sentence in the conclusion section of revised version:
“Using R-GOOSE, communication outside of substation is achievable. This is critical requirements of transmission line protection, communication between substations, and communication between substation to the control center, which are new part of IEC 61850 series. Without routable-GOOSE, GOOSE message can be only exchanged inside the same subnet. This paper deals with feasibility study of R-GOOSE over a public Internet network.”
Reviewer 2 Report
The current reviewed paper deals with the acquisition of external substation data through a public Internet network using Routable GOOSE equipped with a security algorithm that complies with IEC 61850, an international power automation standard, to ensure interoperability. Based on the data acquisition results, a secured Routable GOOSE-based data aggregator was developed and the performance of communication between substations through a public Internet network was analyzed. The following comments need to be improved:
1. What are the research gap and paper contribution compared with others?
2. Improve figures/ table quality such as Table 2 and Figs. 3, 4 and 12.
3. Discussions are poor.
4. Check emergency and congestion events
5. Update the conclusion section with the effective contribution with numerical findings.
English editing needs more attention.
Author Response
Response to Reviewer 2 Comments
First of all, we wish to thank all the reviewers for their constructive comments. These comments helped us to improve the quality of our manuscript. All the issues raised during the review process were addressed in this revised version of the manuscript, hopefully in the most satisfactory way and highlighted in red pen inside the paper. In the following lines, we detail the changes we have made according to the reviewers’ comments.
Point 1: . What are the research gap and paper contribution compared with others?
In this revised version of our manuscript, we add the following sentence in Introduction section:
In this to the best of our knowledge, first relative performance analysis of routable GOOSE on a public Internet. Furthermore, this paper performs a performance comparison of each security algorithm employed, as well as a comparison of system performance that takes into account factors such as CPU load and packet size.
Point 2: Improve figures/ table quality such as Table 2 and Figs. 3, 4 and 12.
In this revised version, we revised the table and figures.
Point 3: Discussions are poor.
In this revised version, we revised the Conclusion section with discussion and the practical benefit of using routable GOOSE.
Point 4: Check emergency and congestion events.
We can get the reviewer’s point. More precise explanation is required.
Point 5: Update the conclusion section with the effective contribution with numerical findings.
In this revised version, we revised the Conclusion section with discussion and the practical benefit of using routable GOOSE.
Reviewer 3 Report
After studying the article, I found the following notes:
1. Figure 3 is not clear.
2. References are not written according to the journal's policy?
3. The introduction needs to be expanded by mentioning related works.
4. What is the contribution of the main paper?
5. The paper lacks a comparative study with published works.
6. What are the goals achieved from this work?
7. What are the pros and cons of Routable GOOSE Security Algorithm?
Author Response
Response to Reviewer 3 Comments
First of all, we wish to thank all the reviewers for their constructive comments. These comments helped us to improve the quality of our manuscript. All the issues raised during the review process were addressed in this revised version of the manuscript, hopefully in the most satisfactory way and highlighted in red pen inside the paper. In the following lines, we detail the changes we have made according to the reviewers’ comments.
Point 1: Figure 3 is not clear.
In this revised version, we revised the table and figures.
Point 2: References are not written according to the journal's policy?
In this revised version, we revised the reference format.
Point 3: The introduction needs to be expanded by mentioning related works.
In this revised version, we revised the Introduction section with the discussion of related works.
Point 4: What is the contribution of the main paper?
As we revised in introduction and conclusion section we add following sentences:
In this to the best of our knowledge, first relative performance analysis of routable GOOSE on a public Internet. Furthermore, this paper performs a performance comparison of each security algorithm employed, as well as a comparison of system performance that takes into account factors such as CPU load and packet size.
Using routable GOOSE, communication outside of substation is achievable. This is critical requirements of transmission line protection, communication between substations, and communication between substation to the control center, which are new part of IEC 61850 series. Without routable-GOOSE, GOOSE message can be exchanged inside the same subnet. This paper deals with feasibility study of routable GOOSE over a public Internet network.
Even with security provision, i.e., AES-128 GCM, average RTT between substation was measured tens of msec range. By performance analysis, we verify that the routable GOOSE satisfies the timing requirements defined in [27]
Point 5: The paper lacks a comparative study with published works.
We add more comparative studies in introduction section.
Point 6: What are the goals achieved from this work?
As we discussed in conclusion as follows, feasibility study of routable GOOSE.
Using R-GOOSE, communication outside of substation is achievable. This is critical requirements of transmission line protection, communication between substations, and communication between substation to the control center, which are new part of IEC 61850 series. Without routable-GOOSE, GOOSE message can be only exchanged inside the same subnet. This paper deals with feasibility study of R-GOOSE over a public Internet network.
Point 7: What are the pros and cons of Routable GOOSE Security Algorithm?
To ensure the confidentiality, security algorithm is critical part by using public Internet. Possible drawback using security algorithm is processing time. We verify total RTT satisfies the timing requirements defined in IEC 61850 90-1 even with security algorithm. We revised the conclusion section.
Reviewer 4 Report
The paper is of interest to specialists who work in non-directional power systems that produce large-scale electricity and supply it using a high-voltage power grid, especially to those who work with current transformer station protection relays and rely only on the data of the internal station it is in to detect changes. Communication technology in terms of data acquisition has become an essential function for state-of-the-art substations. The work is scientifically sound, based on applied research and makes important contributions in the field.
Author Response
First of all, we wish to thank all the reviewers for their constructive comments. These comments helped us to improve the quality of our manuscript. All the issues raised during the review process were addressed in this revised version of the manuscript, hopefully in the most satisfactory way and highlighted in red pen inside the paper. In the following lines, we detail the changes we have made according to the reviewers’ comments.
Round 2
Reviewer 2 Report
The revised version is improved but the following issues needs more attention as:
1. The surevy part should be extended to cover research gap and to assign the accurate paper contribution.
2. Identifying correctly the perfornce indices.
3. Comments must be deep.
4. Update the concultion section with comparison findings and possible future work.
Moderate
Author Response
Response to Reviewer 2 Comments
First of all, we wish to thank all the reviewers for their constructive comments. These comments helped us to improve the quality of our manuscript. All the issues raised during the review process were addressed in this revised version of the manuscript, hopefully in the most satisfactory way and highlighted in red pen inside the paper. In the following lines, we detail the changes we have made according to the reviewers’ comments.
We mark it with underlines win revised version for reviewers’ convenience.
Point 1: The surevy part should be extended to cover research gap and to assign the accurate paper contribution.
We revise the related work parts as follows:
“However, the experiment was conducted on a private network of the FIU testbed and factors such as network hops and CPU load on a public internet network were not taken into account. There was also a study on the definition of R-GOOSE and the application proposal for differences from existing GOOSE messages, power distribution automation, and protection systems [20]. Other studies included Jaya R. A. K. Yellajosula’s imple- mentation of control between substations using R-GOOSE, the allowable range of PURTT (Permissive Under Reach Transfer Trip), and an interlocking application between two substations. However, this study also used a private Internet network, not a public Internet network, and no research on security was conducted [21]. Research on cyber attacks on IEC 61850-based networks was conducted [22], and research on security threats presented in the IEC 62351 standard for message exchange was conducted, but no implementation was conducted to verify communication performance [16]. Communication performance is another important metric in smart grid. Recent research shows that the IoT (Internet of Things) protocols, e.g., DDS and XMPP (Extensible Messaging and Presence Protocol) can meet the timing constraints defined in IEC 61850-90-5 [29,30]. These researches, however, used data model not service model defined in IEC 61850 due to no mapping between IoT protocols and IEC 61850. This research shows that the IEC service model, e.g., R-GOOSE, can be directly in public Internet while satisfying timing constraints.”
Point 2: Identifying correctly the perfornce indices.
We revise the performance analysis part for each indices. We mark it with underlines win revised version for reviewers’ convenience.
Point 3: Comments must be deep.
We add comments for each performance indices. We mark it with underlines win revised version for reviewers’ convenience.
Point 4: Update the concultion section with comparison findings and possible future work.
We revise the conclusion section as follows:
“IEC 61850, the International Standards for power system automation, defines the data and service model. First edition of IEC 61850 defines the GOOSE service for timely delivery of command and data. Due to no mapping of transport layer and network layer, communication inside the same subnet was possible. Thanks to R-GOOSE, communication outside of substation, which means communication between different subnet, is achievable. This is critical requirements for the transmission line protection, communication between substations, and communication between substation to the control center, which are new part of IEC 61850 series. This paper discusses the feasibility study of R-GOOSE over a public Internet network. This study defines the need to acquire data from external substations through the public Internet network and develops a data aggregator that complies with IEC 61850 and acquires data from external substations with security technology. Performance analysis was conducted in an external substation environment using R-GOOSE messages without security technology and R-GOOSE messages with security technology, and RTTs from transmission to transmission were measured. Even with security provision, i.e., AES-128 GCM, average RTT between substation was measured tens of msec range. By performance analysis, we verify that the R-GOOSE satisfies the timing requirements defined in [27]
In addition, since the R-GOOSE messages are transferred using multicast UDP protocol, which has no acknowledgement function, cannot guarantee the delivery of messages which may cause the possibility of loss of data in an environment where abnormalities such as short circuits from external substations must be transmitted or acquired. Therefore, it was confirmed that it is suitable for use in a data aggregator used in a protective relay that relies only on data from internal substations and requires data from external substations such as micro grids located in islands and mountains.
In the future, it is possible to conduct research on the design of external communication with improved security in micro grids environment distributed in islands and mountains using R-GOOSE to which security algorithm is applied, and it is expected to keep pace with changes in the global power market. As the penetration of IoT protocols especially in micro grids, performance comparison between R-GOOSE and IoT protocols, i.e., DDS and XMPP, should be conducted. "
